# Urea-formaldehyde resin room temperature phosphorescent material with ultra-long afterglow and adjustable phosphorescence performance

Organic room-temperature phosphorescence materials have attracted extensive attention, but their development is limited by the stability and processibility. Herein, based on the on-line derivatization strategy, we report the urea-formaldehyde room-temperature phosphorescence materials which are constructed by polycondensation of aromatic diamines with urea and formaldehyde. Excitingly, urea-formaldehyde room-temperature phosphorescence materials achieve phosphor lifetime up to 3326 ms. There may be two ways to enhance phosphorescence performance, one is that the polycondensation of aromatic diamine with urea and formaldehyde promotes spin-orbit coupling, and another is that the imidazole derivatives derived from the condensation of aromatic *o*-diamine with formaldehyde maintains low levels of energy level difference and spin-orbit coupling, thus achieving ultra-long afterglow. Surprisingly, urea-formaldehyde room-temperature phosphorescence materials exhibit tunable phosphorescence emission in electrostatic field. Accordingly, 1,4-phenylenediamine, urea, and formaldehyde are copolymerized and self-assembled into phosphorescence microspheres with different electrostatic potential strengths. By mixing 1 wt% 1,4-phenylenediamine polycondensation microspheres with 1,4-phenylenediamine free microspheres, phosphor lifetime of the composite could be regulated from 27 ms to 123 ms. Moreover, vulcanization process enables precise shaping of urea-formaldehyde room-temperature phosphorescence materials. This work not only demonstrates that urea-formaldehyde room-temperature phosphorescence materials are promising candidates for organic phosphors, but also exhibits the phenomenon of electrostatically regulated phosphorescence.

Recently, organic room-temperature phosphorescent materials (RTP) have received extensive attention due to their luminescent properties, such as long lifetime[1], large Stokes shift[2], and high sensitivity[3]. According to Jablonski diagram[4], promoting intersystem crossover (ISC) process, inhibiting non-radiative transition and reducing triplet exciton quenching are the keys to achieve efficient phosphorescence emission. Consequently, numerous techniques for RTP have been developed, including molecular engineering[5–8], crystal engineering[9,10], host-guest complexation[11–13] and matrix doping strategies[14–18]. Among them, the matrix doping strategy can achieve efficient

✉e-mail: bwwang@tju.edu.cn; lgchen@tju.edu.cn

phosphorescence emission by doping phosphorescent guest molecules in rigid matrix. The RTP materials prepared by matrix doping strategy have the advantages of processing properties and multifunctionality. Indeed, a number of RTP materials achieved excellent RTP performance by doping guest molecules into polymers rich in heteroatomic functional groups, such as polymethyl methacrylate (PMMA)[19,20], polyvinyl alcohol (PVA)[21,22], polyacrylonitrile (PAN)[23,24], hyaluronic acid (HA)[25,26] and polyacrylic acid (PAA)[27,28].

However, the abundant hydrophilic groups in the polymer matrix make it easy to absorb water and oxygen from the surrounding environment, resulting in triplet exciton quenching of the guest phosphor molecule, which reduces the reliability of RTP materials. In order to achieve highly stable pure organic RTP materials, several methods have been developed. Zhao et al.[29] employed a more rigid epoxy resin matrix to effectively isolate $H_2O$ and $O_2$ and inhibit the quenching of triplet excitons, thus achieving stable phosphorescence emission in the air with a phosphorescence decay lifetime ($\tau_p$) of 540 ms. With a difference, our group proposed an on-line derivatization strategy[30]. In the process of copolymerization, guest molecules are derived to form a series of guest molecular derivatives and embedded in the polymer cross-linking network, so as to obtain RTP materials. Accordingly, we successfully developed a RTP material with ultra-long life of 5.33 s. On-line derivatization strategy not only enriching the phosphorus light source of the material, but also the derived phosphor molecules were uniformly and firmly embedded in the matrix of rigid hydrogen bond network, thus improving the stability of triplet excitons and the lifetime of RTP. Therefore, in the construction of RTP materials, the diversification of phosphorescence sources and the shielding of triplet excitons are equally important, and both can jointly promote the long-term stable RTP emission of materials.

Stable ultra-long afterglow is a prerequisite for the application of RTP materials, and the intelligent dynamic response of RTP materials to environmental stimuli further enhances the application value of the RTP materials[31-36]. Actually, for RTP materials obtained through matrix doping strategy, it is hard to achieve dynamic regulation of RTP because of the difficulty in regulating the polarization or orientation of guest molecules in a hindered environment. It has long been proved that the electrostatic fields can change the degree of polarization[37], arrangement and aggregation mode of molecules in a material[38,39], and the photophysical properties of doped RTP materials are undoubtedly closely related to the environment of the guest phosphor molecules. Therefore, we speculated that the polarization and environment of guest molecules can be adjusted by adjusting the electrostatic field intensity, so as to intelligently regulate the RTP emission of materials. Polarization regulation of guest molecules by electrostatic field requires not only appropriate electric field strength but also appropriate gap between molecules and electrodes, which is undoubtedly a challenge in the construction of RTP materials. So far, RTP materials with electric field adjustable phosphorescence have not been reported.

Therefore, a series of urea-formaldehyde room-temperature phosphorescence (UF-RTPs) materials were prepared by the polycondensation of formaldehyde, urea and aromatic diamines based on the on-line derivatization strategy (Fig. 1). Naturally, due to the small amount of aromatic diamine doping, most of urea and formaldehyde polymerized to form urea-formaldehyde resin (UF) as the matrix. The linear polymer chains composed of urea structural unit and methylene group could provide a rigid polar environment for guest phosphor molecules[40]. The doped aromatic diamines could polycondense with formaldehyde and urea on-line to yield urea formaldehyde resin. In addition, the aromatic *o*-diamines could also condense with formaldehyde to generate stable imidazole derivatives embedded in the rigid UF. Thereby, there were multiple phosphorus light sources in UF-RTPs, effectively improving the emission efficiency of phosphors. It is

worth noting that 9,10-diaminophenanthrene doped UF (910DAPT/UF) achieved an ultra-long afterglow up to 47 s and a $\tau_p$ of 3.3 s, and displayed excellent air stability. Furthermore, in order to realize the regulation of RTP performance of materials by electrostatic field, a series of urea-formaldehyde resin RTP microspheres (0%μUF-5%μUF) with different electrostatic potentials were prepared by adjusting the content of *p*-phenylenediamine (14DAP). Among them, 0%μUF displayed the strongest electrostatic potential but no phosphorescence performance, while 1%μUF presented significant phosphorescence emission. The phosphorescence decay lifetime of 1%μUF was increased from 27 ms to 123 ms by blending of 0%μUF with 1%μUF. In addition, UF-RTPs could be easily molded into sheets of different shapes. Due to its good air stability, flexible processability, ultra-long afterglow and dynamically adjustable RTP characteristics, UF-RTPs as pure organic phosphors show a broad application prospect in plastics, coatings, and functional devices.

## Results

### Preparation of UF-RTPs

Three UF-RTPs afterglow materials (14DAP/UF, 13DAP/UF, and 12DAP/UF) were successfully prepared by mixing a guest molecule of 14DAP, *m*-phenylenediamine (13DAP) or *o*-phenylenediamine (12DAP) with paraformaldehyde and then adding it to melted urea (150 °C) for 20 s. 14DAP/UF was taken as an example, and infrared absorption spectrometer (IR) was used to monitor the polycondensation of paraformaldehyde, urea, and 14DAP (Fig. 2a). As the vibration absorption of primary amine (3500 cm⁻¹) decreased, the stretching vibrations of secondary amine (3400 cm⁻¹) and hydroxyl (3100 cm⁻¹) increased, confirming the polycondensation of 14DAP with formaldehyde and urea. By comparing the X-ray diffraction (XRD) patterns of 14DAP/UF and commercial UF (Fig. 2b), the characteristic diffraction peak was observed at 22° for both, but the diffraction peak of 14DAP/UF was sharper, indicating the higher crystallinity of 14DAP/UF. As shown in Supplementary Figs. 8–11, the solid-state nuclear magnetic carbon spectra (SSNMR) of 14DAP/UF, the prepared UF and commercial UF were basically the same, indicating that the UF-RTPs prepared by on-line derivatization strategy maintained the structure of UF resin. Notably, the relative peak intensity of carbon of methylene (48 ppm) in 14DAP/UF SSNMR spectrum (Supplementary Fig. 9) is higher than that of commercial UF (Supplementary Fig. 11), suggesting that UF-RTPs were mainly composed of linear UF molecules. The higher the content of linear molecules, the better the crystal shape of the polymer, which is consistent with the results of XRD spectra. Furthermore, molecular weight analysis of 14DAP/UF and commercial UF was conducted using gel permeation chromatography (GPC). The results revealed that the molecular weight of 14DAP/UF (Supplementary Fig. 12) was 1040 g/mol, which was lower than that of commercial UF (2112 g/mol, Supplementary Fig. 13). X-ray photoelectron (XPS) spectra showed only C, O, and N elements existed in 14DAP/UF, indicating that the pure organic RTP material was successfully prepared (Supplementary Fig. 14).

### Photophysical properties of UF-RTPs

With the UF-RTPs in hand, we further investigated their photophysical properties. As shown in Fig. 2c, 14DAP/UF, 13DAP/UF, and 12DAP/UF all exhibited photoluminescence property (Supplementary Movie 1), producing blue emission under 310 nm UV excitation. After ceasing irradiation, the three showed obvious dark blue to light blue afterglow emission. In order to investigate the photophysical properties of UF-RTPs, the steady-state photoluminescence (PL) spectra and delayed photoluminescence (P) spectra of the three UF-RTPs were recorded. The peak of 298 K phosphorescence spectra of 14DAP/UF and 13DAP/UF (Fig. 2c) located at 451 nm and 432 nm, respectively. Compared with the corresponding 298 K steady-state PL spectrum, there is a significant redshift, showing significant phosphorescence emission.

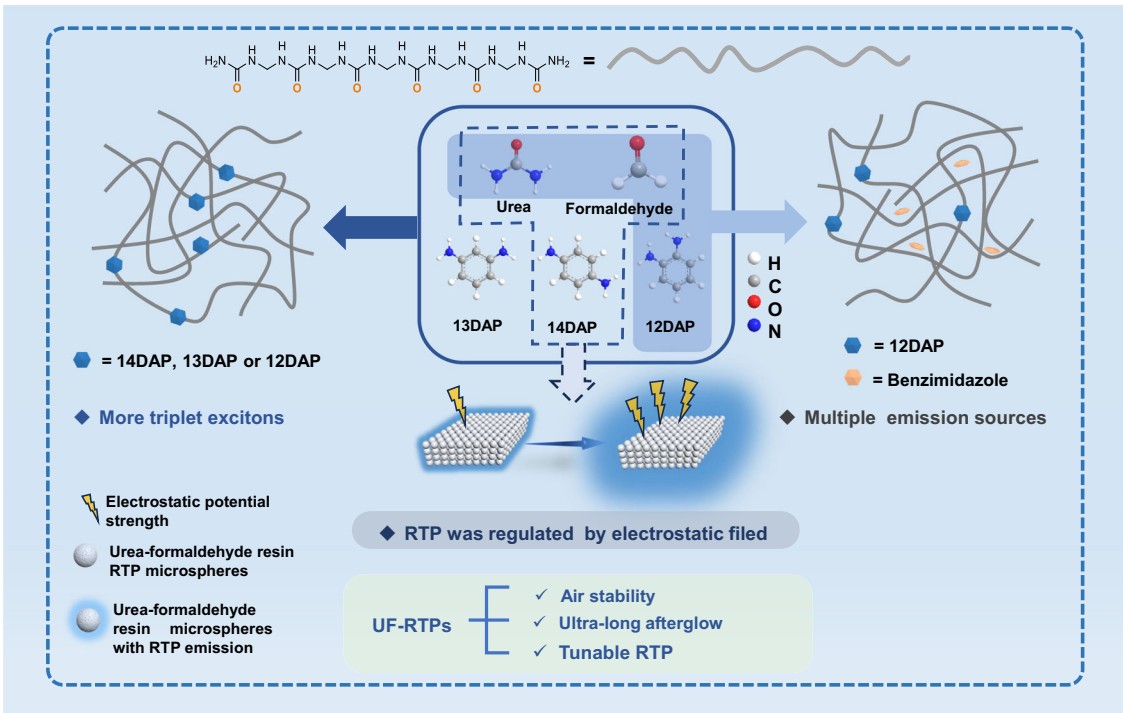

**Fig. 1 | Schematic diagram of the mechanism of the on-line derivatization.** Efficient phosphorescence emission and dynamically tunable phosphorescence of UF-RTPs achieved based on on-line derivatization strategy.

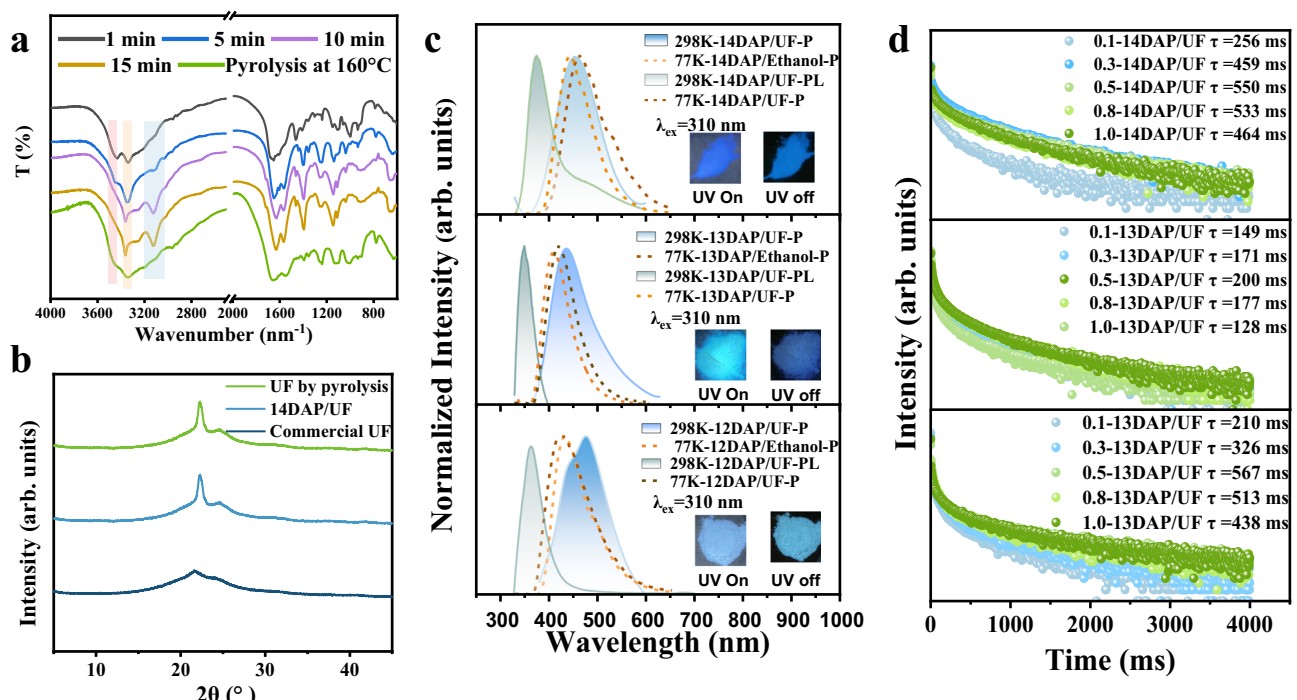

**Fig. 2 | Structural and Photophysical properties characterization of UF-RTPs.**
**a** IR spectra monitoring polycondensation; **b** XRD patterns of commercial UF, 14DAP/UF and prepared UF. **c** 298 K steady-state PL spectra, 298 K phosphorescence spectra (delay 1 ms), 77 K phosphorescence spectra (delay 1 ms), and 77 K dilute solution phosphorescence spectra of 14DAP/UF, 13DAP/UF, and 12DAP/UF; **d** Phosphorescence decay lifetime of UF-RTPs with dopant ratio of guest molecule from 0.1wt%-1wt%.

What's more, two emission peaks at 443 nm and 472 nm were observed at 12DAP/UF phosphorescence spectrum, which may be because in the polycondensation process, 12DAP not only condensed with urea and formaldehyde, but also reacted with formaldehyde to form a new phosphorus source. In order to confirm the phosphorescent centers of

UF-RTPs, 77k phosphorescence spectra of guest molecules and UF-RTPs were collected, respectively (Fig. 2c). 77K-14DAP/UF-P (447 nm) and 77K-14DAP/Ethanol-P (461 nm) showed a significant overlap. 77K-13DAP/UF-P (413 nm) and 77K-13DAP/Ethanol-P (419 nm) also showed the same result. Thus, the luminescence centers of 14DAP/UF and

13DAP/UF were 14DAP and 13DAP, respectively. However, 77K-12DAP/UF-P (429 nm) and 77K-12DAP/Ethanol-P (430 nm) are consistent with the peak at 432 nm in 298K-12DAP/UF-PL, indicating that the peak at 432 nm belongs to 12DAP. Furthermore, considering the influence of the guest molecule doping on RTP performance, a series of UF-RTPs with different guest molecule doping amounts were prepared and their RTP performance were investigated (Supplementary Fig. 15). At a doping ratio of 0.5 wt%, 14DAP/UF and 13DAP/UF achieved the longest $\tau_p$ of 550 ms and 200 ms, respectively, while 12DAP/UF achieved the longest afterglow of 7.5 s with $\tau_p$ of 567 ms (Fig. 2d). And the photoluminescent quantum yields ($\Phi_{pl}$) of 14DAP/UF, 13DAP/UF, and 12DAP/UF reached 5.41%, 3.53% and 4.37%, respectively (Supplementary Fig. 23a–c). The phosphorescent quantum yields ($\Phi_p$) of 14DAP/UF, 13DAP/UF, and 12DAP/UF reached 1.56%, 1.28%, and 0.89%, respectively. These results fully manifested that the on-line derivation of aromatic diamine and embedding into rigid UF is a promising strategy for constructing room temperature phosphorescent materials.

To further confirm the structure of the luminescent sources in UF-RTP, the oligomers and guest molecular derivatives in UF-RTP were extracted with alcohol and analyzed by LC-MS (Supplementary Figs. 16–18). A series of copolymerized derivatives of aromatic diamine with urea and formaldehyde were detected in the corresponding extracts (Supplementary Table 1), while benzimidazole (BMZ) ($[M + 1]^+ = 119.1$) was detected only in 12DAP/UF (Supplementary Fig. 18), matching the phosphorescence spectrum of 298K-12DAP/UF-P (Fig. 2c). It is not difficult to see that there are two derivation pathways of aromatic diamine in the polycondensation process, one is the polycondensation of 12DAP with urea and formaldehyde, and the other is the condensation of 12DAP with formaldehyde to form BMZ. In conclusion, 12DAP/UF exhibited the best phosphorescence performance, which can be attributed to that the on-line derived BMZ markedly improved the phosphorescence decay lifetime of 12DAP/UF.

## Phosphorescence mechanism of UF-RTPs

To gain insight into the luminescence mechanism of 12DAP/UF with multiple emission sources, molecular dynamics simulation (MDS), electrostatic potential (ESP), and binding energy (BE) were calculated based on density functional theory (DFT). And the initial and final atomic coordinates of the molecular structures for calculation of ESP or BE were shown in Supplementary Data 1 and 2, respectively. As shown in Supplementary Fig. 19, the molecular spatial configuration of urea-formaldehyde resin was analyzed by MDS, and it was found that the molecular chain of urea-formaldehyde resin would fold and bend under electrostatic forces with the growth of polymer chain. It is not difficult to see that such UF molecular spatial configuration is conducive to inhibiting the migration of guest molecules. In addition, the calculation results of ESP indicated that UF resin molecules contain both positive and negative potential centers, with oxygen atoms as negative centers and nitrogen atoms as positive centers (Supplementary Fig. 19). Moreover, a series of diaminobenzene derivatives were derived on-line during the polycondensation of diaminobenzene, urea and formaldehyde. Using 14DAP-UF, 13DAP-UF and 12DAP-UF as examples, ESP analysis (Fig. 3a) showed that this derivatization mechanism effectively increased the number of ESP positive and negative centers in RTP materials (Supplementary Tables 2–4). Diaminobenzene derivatives were able to form stronger electrostatic interactions with UF through reversed ESP, resulting in a higher binding energy between the matrix and guest molecules, about twice that of diaminobenzene (Fig. 3b). The matrix itself could form chain clusters through the electrostatic interaction between molecular chains, which is conducive to limiting the vibration and migration of guest molecules. Therefore, the on-line derivatization of diaminobenzene could effectively inhibit the non-radiative transition of guest molecules, thus greatly improving the RTP performance of UF-RTPs.

Based on the two derivatization pathways of diaminobenzene in UF, the energy levels, electron conversion characteristics, and spin-orbit coupling (SOC) of 14DAP, 13DAP, 12DAP, 14DAP-UF, 13DAP-UF, 12DAP-UF and benzimidazole (BMZ) were calculated by DFT. In the polycondensation of aromatic diamine with urea and formaldehyde, the aromatic diamine as a guest fluorescent molecule was embedded in the polymer chain, which not only effectively enhanced SOC, but also significantly inhibited non-radiative transition (Supplementary Fig. 20). In addition, BMZ derived from 12DAP and formaldehyde during the polycondensation was another phosphor source with a very small energy level difference between $S_1$ and $T_4$ ($\Delta E_{(S1-T4)}$) of only −0.005 eV and low levels of SOC (Supplementary Fig. 20g), showing another mechanism of ultra-long phosphorescence emission in the UF-RTPs. Thus, the two phosphor sources derived from aromatic diamine in UF correspond to two phosphor emission mechanisms (Fig. 3c). The polycondensation of aromatic diamine with urea and formaldehyde increased its SOC and the phosphorescence emission efficiency, while BMZ derived from 12DAP and formaldehyde with low levels of $\Delta E_{ST}$ and SOC promoted the stability of the triplet excitons and prolonged the phosphorescence decay life.

## Other aromatic diamines for UF-RTPs

Based on the excellent RTP performance of diaminobenzene doped UF, a series of aromatic diamines such as 2,3-diaminonaphthalene (23DAN), 1,8-diaminonaphthalene (18DAN), 1,5-diaminonaphthalene (15DAN), 1,4-diaminonaphthalene (14DAN), and 9,10-diaminophenanthrene (910DAPT) were further selected to construct UF-RTPs (Supplementary Movie 2). Excitingly, all UF-RTPs exhibited long afterglow emission (Supplementary Fig. 21), where 910DAPT achieved the longest blue afterglow of 47 s (Supplementary Movie 3) with $\tau_p$ of 3326 ms and $\Phi_p$ of 11.85% (Fig. 4a, b), and 23DAN/UF also achieved the longest green afterglow of 15 s with $\tau_p$ of 932 ms and $\Phi_p$ of 5.14%. 18DAN/UF, 15DAN/UF, and 14DAN/UF also achieved ultra-long $\tau_p$ of 712 ms, 355 ms, and 333 ms, respectively (Fig. 4b). In fact, ultra-long organic RTP achieved by doping aromatic diamines in urea-formaldehyde resins was rare among the reported halogen-free organic RTP materials, and the on-line derivatization strategy of aromatic diamine during the polycondensation of aromatic diamine with formaldehyde and urea provides a promising platform for the development of ultra-long afterglow materials.

Similarly, 1H-naphtho[2,3-d]imidazole (NMZ) and 1H-phenanthro[9,10-d]imidazole (PMZ) were detected in 23DAN/UF (Supplementary Fig. 25) and 910DAPT/UF (Supplementary Fig. 26), respectively. The luminescence mechanism of imidazole phosphors such as BMZ, NMZ, and PMZ in UF was further studied because of their excellent RTP performance. For comparison, BMZ, NMZ, and PMZ were directly doped into UF to afford BMZ/UF, NMZ/UF, and PMZ/UF, respectively. The phosphorescence spectra of 910DAPT/UF and PMZ/UF were compared, and there is a significant overlap. In addition, 12DAP/UF and BMZ/UF, 23DAN/UF and NMZ/UF showed the same results, respectively. It confirmed that imidazole derivatives were the aromatic o-diamine/UF phosphor sources. The $\Delta E_{(S1-T3)}$ of NMZ and $\Delta E_{(S1-T4)}$ of PMZ were calculated to be −0.058 eV and 0.076 eV respectively, and the two energy level differences were quite small (Supplementary Fig. 28b, c), so the SOC of both NMZ and PMZ was kept at a very low level, which is consistent with the energy level structure of BMZ. Therefore, imidazole derivatives as phosphor light sources, including BMZ, NMZ, and PMZ, exhibited consistent ultra-long phosphorescence mechanisms due to their lower $\Delta E_{ST}$ and SOC.

Stability was also the key to the wide application of RTP materials. Firstly, the phosphor emission intensity and phosphor decay lifetime of UF-RTPs in $N_2$ and air were compared (Supplementary Fig. 29). The results showed that the phosphorescence intensity in air was slightly lower than that of UF-RTPs in $N_2$. Nevertheless, it still maintained strong phosphor emission intensity and good phosphor decay life

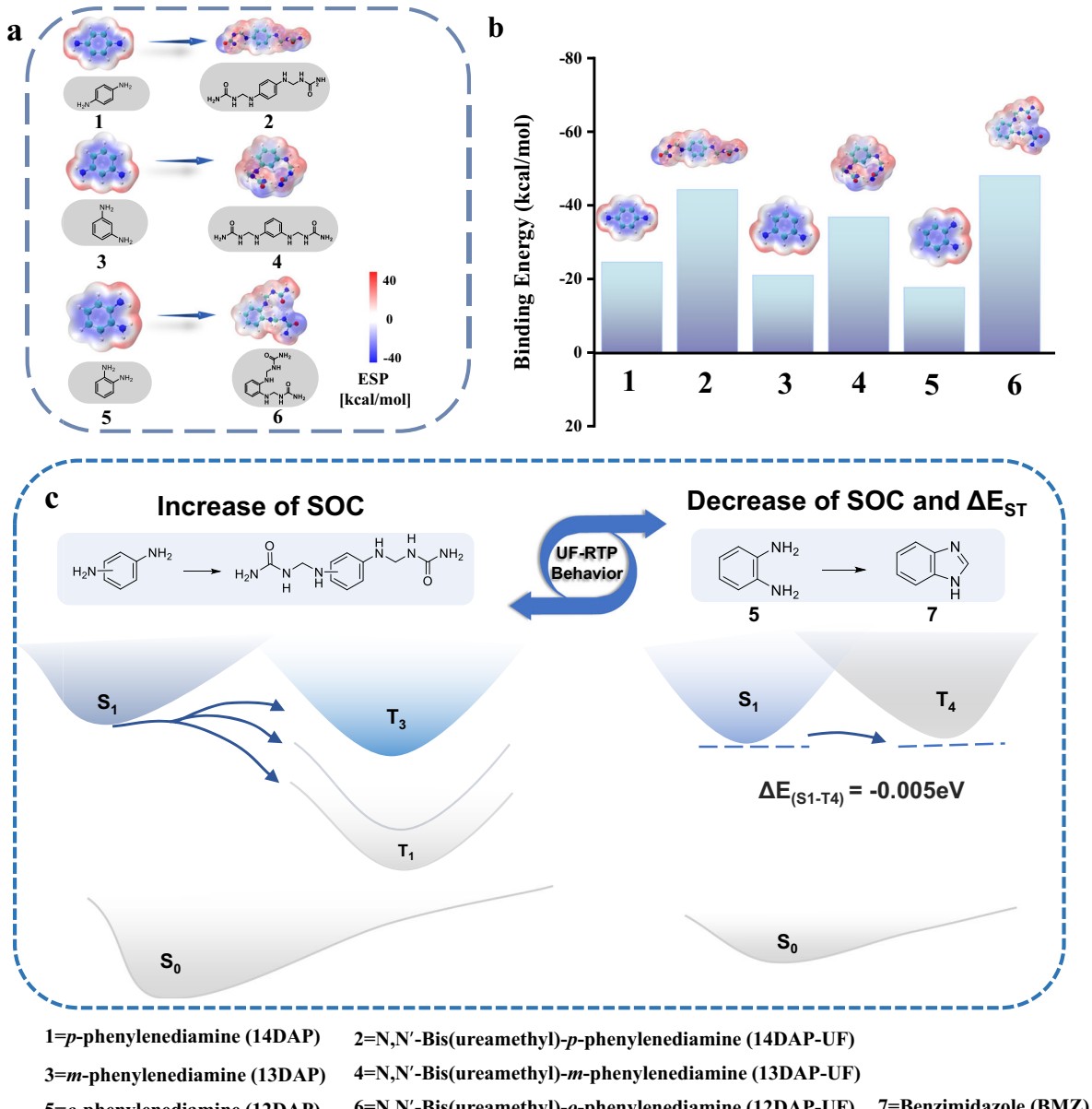

**Fig. 3 | Quantum chemical calculations on matrix, guest molecules and the derivatives.** ESP distribution of (**a**) the guest molecules and their derivatives; (**b**) binding energy of guest molecules and their derivatives to UF-5 (Supplementary Fig. 19b); (**c**) two phosphorescence enhancement mechanisms of UF-RTPs.

1=*p*-phenylenediamine (14DAP)     2=N,N′-Bis(ureamethyl)-*p*-phenylenediamine (14DAP-UF)

3=*m*-phenylenediamine (13DAP)     4=N,N′-Bis(ureamethyl)-*m*-phenylenediamine (13DAP-UF)

5=*o*-phenylenediamine (12DAP)     6=N,N′-Bis(ureamethyl)-*o*-phenylenediamine (12DAP-UF)     7=Benzimidazole (BMZ)

(Supplementary Fig. 30). Furthermore, the phosphorescence decay spectra of 910DAPT/UF (Fig. 4c) showed that its excellent afterglow performance was still maintained after 6 months of exposure to air and sunlight. In addition, 910DAPT/UF maintained a good phosphor lifetime during four cycles of washing and drying (Supplementary Fig. 31). These results indicated that UF-RTPs exhibited excellent stability.

## Construction of microspheres for tunable RTP by electrostatic field

The successful construction of stable ultra-long RTP materials encouraged us to further attempt to regulate the RTP performance of UF-RTPs. Firstly, the feasibility of the above idea was verified by constructing urea-formaldehyde resin microspheres (μUFs) with different electrostatic potential strengths. Thus, a series of μUFs (0%μUF, 0.5% μUF, 1%μUF, 2%μUF and 5%μUF) were synthesized by polycondensation and self-assembly of urea, formaldehyde, and a certain amount of 14DAP (Fig. 5a). Scanning electron microscopy (SEM) images showed that μUFs were successfully prepared (Supplementary Fig. 3) with

uniform particle size of about 1–2 μm (Supplementary Fig. 34). Then, the electrical characteristics of μUFs were further investigated. With the increase of 14DAP mass ratio, the dielectric constant of 0%μUFs to 5%μUFs continuously decreased at the high frequency part (Supplementary Fig. 37a), indicating that the overall polarizability of μUFs was decreased with the increase of 14DAP content. In addition, the electrochemical impedance of 0%μUFs to 5%μUFs was continuously reduced (Fig. 5e), further indicating that doping of 14DAP can enhance the conductivity of μUFs and reduce the accumulation of static charge on the surface of μUF. These results revealed that the electrostatic potential and electrical property of μUFs can be effectively regulated by changing the doping ratio of 14DAP.

In a further attempt to tune the RTP performance of μUF by electrostatic field, their PL spectra and phosphorescence spectra were collected (Supplementary Fig. 35), and the $\tau_p$ of 0% μUF, 0.5% μUF, 1% μUF, 2% μUF, and 5% μUF were 3 ms, 13 ms, 27 ms, 21 ms, and 5 ms, respectively (Supplementary Fig. 36, Supplementary Movie 4). Considering the significant differences between 0% μUF and 1% μUF in

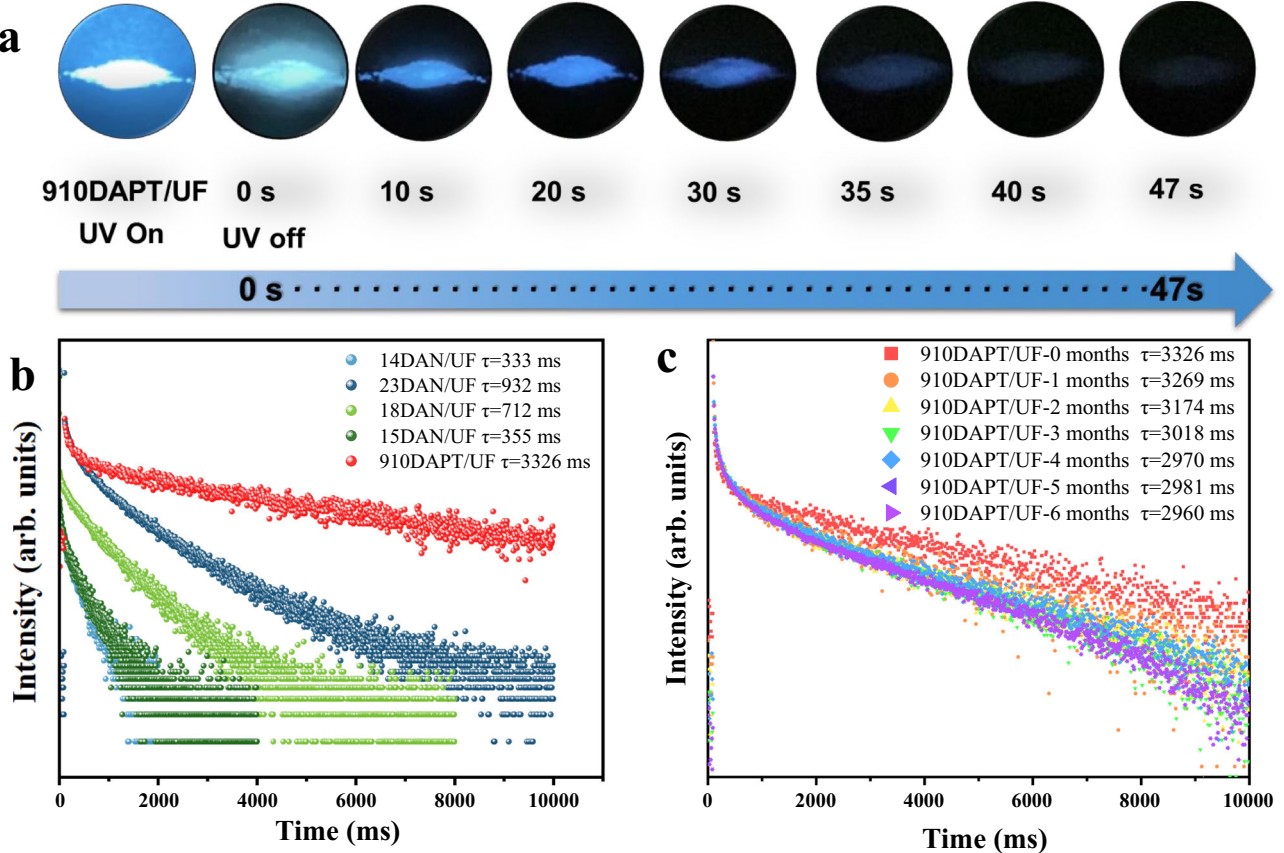

**Fig. 4 | Photophysical properties of UF-RTPs. a** 910DAPT/UF long afterglow; (**b**) DAN/UFs and 910DAPT/UF phosphorescence decay spectra ; (**c**) phosphorescence decay spectra of 910DAPT/UF exposed to air at different stages.

conductivity and afterglow performance, 0% μUF and 1% μUF were chosen for further study. Subsequently, different proportions of 0% μUF and 1% μUF were directly mixed in a glass bottle, and the changes in the RTP performance of the mixture were observed (Fig. 5b). When $m_{0\%\mu UF}$: $m_{1\%\mu UF}$ increased from 0 to 2, the $\tau_p$ of the mixture increased from 27 ms to 123 ms (Fig. 5f). The electrostatic regulation mechanism of RTP was further studied by characterizing the electrical properties of mixtures of 0% μUF and 1% μUF. With the increase of 0% μUF content (Supplementary Fig. 37b), the overall dielectric constant of the mixture was effectively enhanced, which facilitated the accumulation of the static charge. The electrostatic field strength of 0% μUF and 1% μUF were determined by electrostatic force microscopy (EFM). The surface probe amplitudes of 0% μUF and 1% μUF were 875 and 271 mdeg (Fig. 5c, d), respectively, indicating that the intensity of surface electrostatic potential of 0% μUF was nearly 3 times that of 1% μUF (Supplementary Movie 5). Furthermore, the surface potential of 0% μUF and 1% μUF were 670 nV and 256 nV, respectively in the Kelvin Probe Force Microscope (KPFM) potential profile (Supplementary Fig. 39). The surface potential of 0% μUF was 2.6 times that of 1% μUF, indicating that doping of 14DAP reduced the surface potential of μUF, which was consistent with the dielectric constant of μUF (Supplementary Fig. 37a). Moreover, in the KPFM potential diagram of 0% μUF and 1% μUF mixtures (Supplementary Fig. 39f), the cluster behavior of two kinds of microspheres with different surface potentials could be clearly observed.

Therefore, it is speculated that the transfer of static charge from 0% μUF to 1% μUF enhanced the dielectric property of 1% μUF and the polarization of guest molecules, thus improving its RTP performance. This intriguing RTP mechanism in response to electrostatic field provides a way to regulate the photophysical properties of phosphorescent materials, and is helpful to further explore the regulatory

mechanism of phosphorescence, and develop electrostatic field chemical sensors and pure organic phosphorescent devices.

## UF-RTPs powder for molding

Phosphor powder is commonly used in coatings[41], plastics[42], and toys. The commercial phosphors are mainly derived from rare earth metal complexes, which will inevitably cause harm to the environment and human health in the processing process and use. Pure organic phosphors are more beneficial to human life due to abundant raw materials, low toxicity, and minimal pollution. Based on their good processing and phosphorescent properties, UF-RTPs powder was filled into the mold and kept at 140 °C and 20 MPa for 40 s (Fig. 6a) to afford UF-RTP sheets of different shapes (Fig. 6b–d). 910DAPT/UF, 23DAN/UF, and 14DAP/UF powders were processed into shields, and their ultra-long afterglows of 31 s, 9 s, and 6 s were observed under 310 nm UV excitation (Fig. 6e–g, Supplementary Movie 6). In summary, UF-RTPs are promising pure organic RTP powders, which are convenient for preparation, mass production and molding, and good stability.

## Discussion

In this work, a series of ultra-long afterglow UF-RTPs were constructed based on the on-line derivatization strategy, which exhibited excellent air stability, processability and dynamic tunability. Among them, 910DAPT/UF achieved an ultra-long afterglow of up to 47 s and a $\tau_p$ of 3.3 s, with no obvious RTP performance attenuation in 6 months under environmental conditions. IR, XRD, SSNMR, and mass spectra showed that there were two derivatization pathways of aromatic diamines in the preparation of UF-RTPs. First, the guest fluorescent molecule was embedded into the polymer chain through the polycondensation of aromatic diamine with urea and formaldehyde, which enhanced the SOC of the guest molecule and improved the phosphor emission

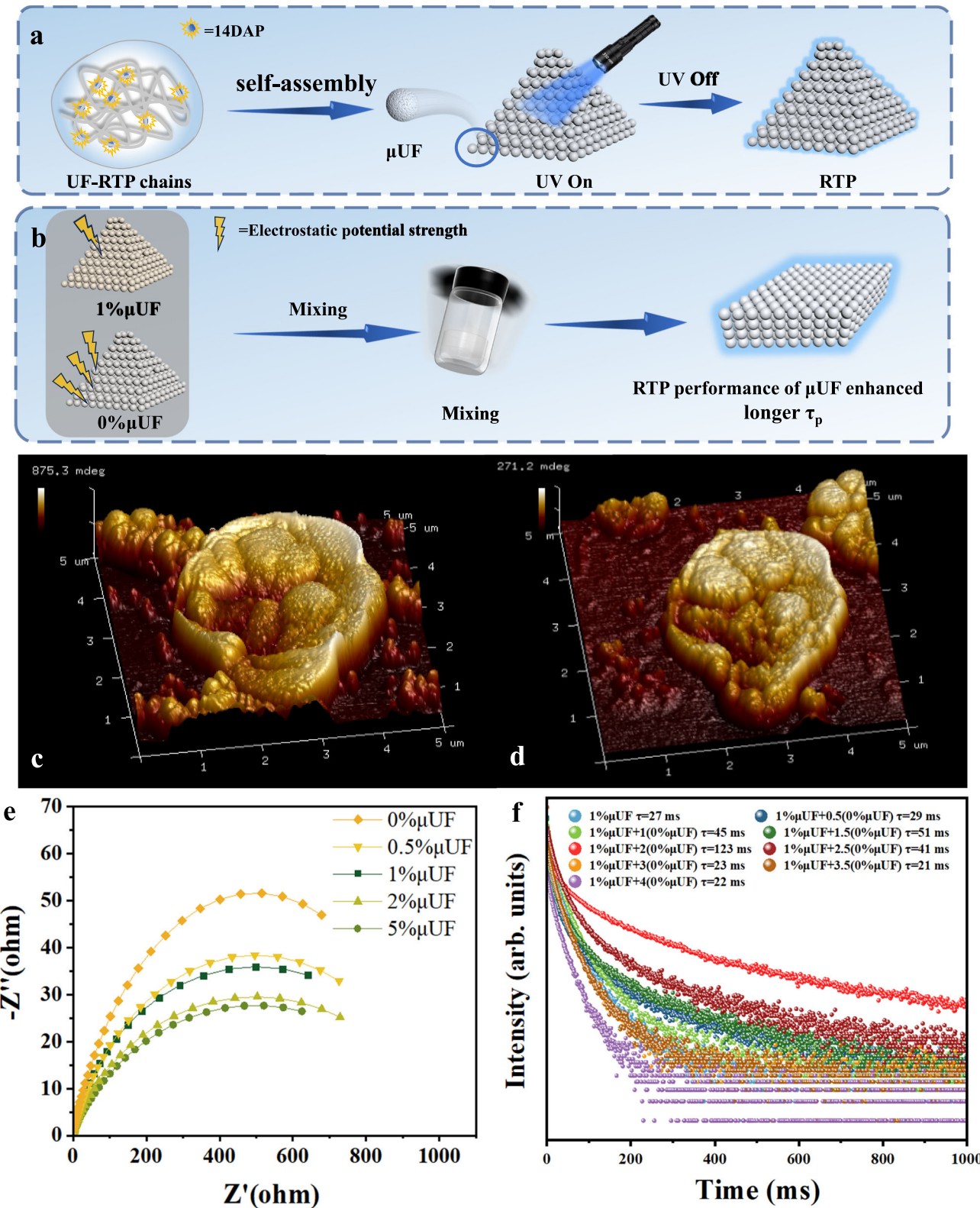

**Fig. 5 | The construction process, electrical properties, and photophysical characteristics of μUFs. a** μUFs preparation diagram; (**b**) The composite of 0% μUF and 1% μUF with adjustable RTP performance; EFM scanning 3D images of (**c**) 0% μUF and (**d**) 1% μUF; (**e**) electrochemical impedance testing of 0% μUF-5% μUF; (**f**) phosphorescence decay spectra of the composites with different mass ratio of 0% μUF and 1% μUF.

efficiency. Second, the imidazole derivatives (BMZ, NMZ, and PMZ) generated on-line from *o*-aromatic diamine and formaldehyde had low levels of $\Delta E_{ST}$ and SOC, and could be in situ embedded in rigid UF-RTPs to achieve ultra-long afterglow emission. Furthermore, theoretical

calculations also supported that the derivatization strategy of aromatic diamine could improve the RTP emission efficiency. Through the vulcanization process, UF-RTPs could be easily molded into sheets of different shapes. Notably, urea, formaldehyde and different amounts

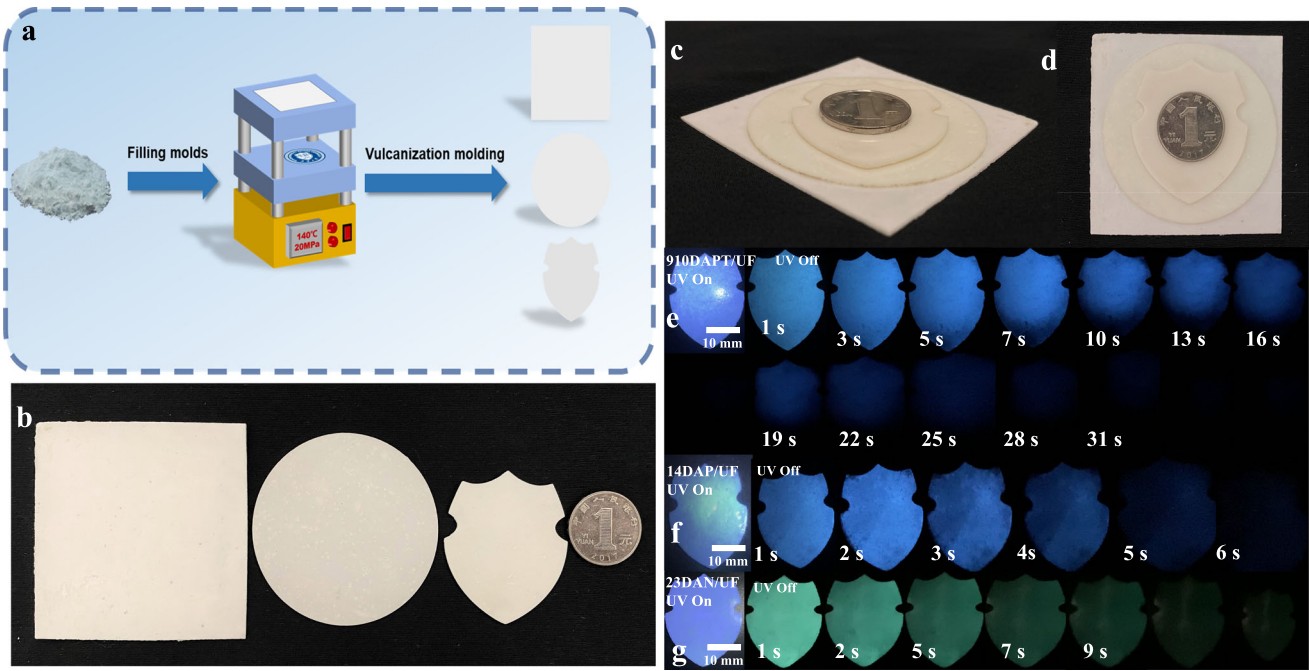

**Fig. 6 | UF-RTPs sheets and afterglow performance. a** Processing diagram of UF-RTPs sheets using plate vulcanization; (**b**) front view, (**c**) side view, and (**d**) top view of square, circular, and shield shaped UF-RTP plates; (**e**) ultra-long afterglow of 910DAPT/UF, (**f**) 14DAP/UF, (**g**) 23DAN/UF shield plates.

of 14DAP underwent polycondensation and self-assembled into μUFs with different electrostatic potentials. By mixing 0%μUF and 1%μUF to change the electrostatic potential, the mixture achieved intriguing electrostatic adjustable RTP performance, and $\tau_p$ varied from 27 ms to 123 ms. Therefore, the on-line derivation strategy provides an effective way to achieve efficient RTP and dynamically tunable RTP materials, which will expand the application of RTP materials in plastics, coatings, electrostatic field chemical sensor and functional devices.

## Methods
### Materials and methods
All chemicals were purchased from commercial sources without further purification. Paraformaldehyde was purchased from Tianjin Chemical Reagent Research Institute Co. Ltd. 1,2-Diaminobenzene (purity: 99%) and 9,10-diaminophenanthrene (purity: 98%) were obtained from Shanghai Macklin Biochemical Technology Co. 40% Formaldehyde solution (analytic reagent) and urea (purity: 99%) were purchased from Jiangtian Technology Co. LTD. 1,3-Diaminobenzene (purity: 98%), 2,3-diaminonaphthalene (purity: 98%) and 1,8-diaminonaphthalene (purity: 99%) were obtained from Shanghai Meryer Biochemical Technology Co., LTD. 1,4-Diaminonaphthalene (purity: 98%) was purchased from Shanghai New Platinum Chemicals Technology Co. LTD.

Steady-state spectra, delayed PL spectra, and time-resolved emission decay curves were performed on an Edinburg FLS1000 fluorescence spectrophotometer (Edinburgh Instruments, Britain). Infrared absorption spectra were measured on a Nicolet 380 FT-IR spectrometer. X-ray diffraction (XRD) analyses were carried out on AXS D8 X-ray diffractometer (Bruker, America) using a Cu Kα X-ray source (40 kV, 100 mA). HRMS were measured by a MicroTOF-Q II (Bruker, America). X-ray Photoelectron Spectroscopy (XPS) were performed using a Kalpha (Thermo, America). Kelvin Probe Force Microscope (KPFM) and Electrostatic force microscope (EFM) spectra were collected by Bruker Dimension Atomic Force Microscope. Dielectric properties were tested by Agilent 4294 A. Electrochemical properties of the material frameworks were measured through a three-electrode system in an electrochemical workstation with a brand of CHI66 (Chenhua, China). Solid state NMR (SSNMR) was performed using the

Avance Neo 400WB (Bruker, America). Gel permeation chromatography (GPC) was performed by 1260 Infinity II (Agilent, America).

### Preparation of UF-RTPs
Taking 14DAP/UF as an example, 2.0 g urea was added to 100 mL one necked flask and heated at 150 °C until melted. Then, 15.0 mg 14DAP was mixed with 1.0 g paraformaldehyde powder, and added to the melted urea. The mixture was stirred at 150 °C for another 20–30 s to become a solid, then cooled to room temperature, collected, dried, and ground to yield UF-RTPs powder. Similarly, 12DAP/UF, 13DAP/UF, 23DAN/UF, 14DAN/UF, 14DAN/UF, 18DAN/UF and 910DAP/UF materials were prepared.

### Preparation of urea-formaldehyde resin microspheres (μUF)
Urea-formaldehyde resin microspheres were prepared by dispersion polymerization. 1.5 g urea, 4.1 g 40% formaldehyde aqueous solution, 30.0 mg 14DAP, and 50 mL water were added into a one-neck flask. After dissolution, 50.0 mg hydroxymethyl cellulose and 4.0 g ammonium sulfate were added as dispersant and precipitant, respectively. Then, the pH of the reaction mixture was adjusted to 2, heated to 50 °C for 3 h, cooled to room temperature and aged for 24 h, filtered and collected precipitates, centrifuged and washed with water for 5 times, and dried at 50 °C for 3 h to afford μUFs.

### Forming and processing of UF-RTPs sheets
The flat vulcanizing machine used in the forming process of UF-RTPs sheets is 20 T from Baoding Precision Instrument Co., Ltd. The UF-RTPs sheets were processed at 140 °C and 20 MPa for 60 s.

## Data availability
All the other data used in this study are available in the article and its supplementary information files and from the corresponding author upon request.

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

## Acknowledgements

Thanks to the Instrumentation Platform of the School of Chemical Engineering and Technology, Tianjin University, for their support and assistance in the analysis of compound structures and characterization of material dielectric properties.

## Author contributions

Wensheng Xu: conceptualization, investigation, formal analysis, validation, writing-original draft, writing-review & editing. Bowei Wang: supervision, project administration, writing - review & editing. Shuai Liu: methodology, formal analysis. wangwang fang: data curation, formal analysis. qinglong jia: data curation, formal analysis. Jiayi Liu: data curation, formal analysis. Changchang Bo: data curation, formal analysis. Xilong Yan: data curation. Yang Li: methodology. Ligong Chen: supervision, project administration, writing - review & editing.

## Competing interests

The authors declare no competing interests.

## Additional information

Wensheng Xu[1], Bowei Wang ®[1,2,3,4] ✉, Shuai Liu[5], Wangwang Fang[1,2,5], Qinglong Jia[1], Jiayi Liu[1], Changchang Bo[1], Xilong Yan[1,2,3,4], Yang Li[1,3] & Ligong Chen ®[1,2,3,4] ✉

[1]School of Chemical Engineering and Technology, Tianjin University, Tianjin 300350, People's Republic of China. [2]Zhejiang Institute of Tianjin University, Shaoxing 312300, PR China. [3]Collaborative Innovation Center of Chemical Science and Engineering (Tianjin), Tianjin 300072, PR China. [4]Tianjin Engineering Research Center of Functional Fine Chemicals, Tianjin 300350, PR China. [5]Shaoxing Xingxin New Materials Co., Ltd, Shaoxing, Zhejiang, PR China. ✉e-mail: bwwang@tju.edu.cn; lgchen@tju.edu.cn

