## [Peer Review File · Nature Communications]

Urea-formaldehyde resin room temperature phosphorescent material with ultra-long afterglow and adjustable phosphorescence performanceREVIEWER COMMENTS

Reviewer #1 (Remarks to the Author):

In this work, based on the online derivative strategy proposed by the authors, they developed a urea-formaldehyde resin RTP material with ultra-long afterglow, good stability and excellent processability. The phosphorescence mechanism of UF-RTPs was also investigated. Interestingly, electrostatic field regulation of RTP is achieved by simply mixing RTP microspheres with different electrostatic strengths. The results in this manuscript are of great interest to readers in this field. Therefore, the manuscript could be published after minor revisions.

1.The drawing in Figure 3c is not standard, and the arrow pointing should be clear.

2.Figure 4b should add the label of phosphorescence decay lifetime.

3.Why use 310 nm UV as excitation source in the photophysical test of UF-RTPs? “As shown in Fig. 2c, 14DAP/UF, 13DAP/UF, and 12DAP/UF all exhibited photoluminescence property, producing blue emission under 310 nm UV excitation.”

4.UF-RTPs prepared by pyrolysis showed two efficient phosphorescence emission mechanisms. Is there any synergistic effect between them?

5.Phosphorescence induced by electrostatic interaction between different microspheres is a very interesting method for enhancing RTP. The adjustment performance is achieved by mixing microspheres with different electrostatic strengths. However, can directly applied electrostatic fields regulate phosphorescence properties on RTP microspheres? “When $m_0\% \mu\text{UF}$: $m_1\% \mu\text{UF}$ increased from 0 to 2, the τ_p of the mixture increased from 27 ms to 123 ms (Fig. 5f).”

6.some typical literature about long-lived RTP should be cited, such as 10.1021/jacs.2c02076, 10.1002/sml.202201223, 10.1002/ange.202203254, 10.1002/adma.202204415, 10.1038/s41467-023-43133-1, 10.1016/j.chempr.2023.05.023, etc.

Reviewer #2 (Remarks to the Author):

In this paper, Chen et al. reported a series of ultra-long afterglow UF-RTPs constructed based on the on-line derivatization strategy, which show a series of attractive properties. It should note that “the polymerization of aromatic diamines with urea and formaldehyde” is NOT a copolymerization, conceptually. In addition, the characterization of the samples are insufficient, only IR spectra presented. The monomer residues in the final polymers should be given, and the self-assembly to microspheres should be characterized, and provides reasonable explanations. Overall, this paper could be accepted once the following issues are well addressed.

Major revisions:

1)In Figure 1, The benzimidazole is exhibited in the polymer chain by physical blending not chemical bond connections, so what would be the difference if the benzimidazole molecules were directly physically doped? In addition, this fig is full of many phrases, which one is the most important?

- 2)The characterization of UF-RTPs' structure needs to be further clarified, e.g. NMR results and molecular weight data need to be given.
 - 3)In the photophysical properties of UF-RTPs part, figure 2c, the authors need to further explain why the phosphorescence wavelengths are different in different temperatures, and why there are also two emission peaks in the phosphorescence spectra of 298K-14DAP/UF-P?
 - 4)the authors need to further explain in Figure 2d, why the phosphorescence lifetime of UF-RTPs is first increased and then decreased with the dopant ratio of guest molecule from 0.1wt%-1wt%?
 - 5)In Figure 4c, The authors demonstrate the phosphorescent lifetime of the polymer after 6 months of storage, when the the polymer is placed in water and then dried, how does the phosphorescent lifetime change?
 - 6)Apart from the phosphorescent lifetime, the quantum yield of phosphorescence and fluorescence needs to be given, and the difference in the strength and lifetimes of the polymer in noble gas and air should be further compared.
 - 7)More experiments and evidence are needed to elucidate the effect of dielectric property of the mixture and the polarization of guest molecules to its RTP performance. And how to explain the decreased lifetime with the higher mass ratio of 0% μ UF and 1% μ UF.
- Minor revisions:
- 8)Carefully check the whole manuscript. There are some mistakes, for example, Ref. 28 citations are formatted incorrectly; RNH3 and R2NH2 are the wrong expressions.
 - 9)In Figure 1, there is no appearance change shown in the schematic diagram of the ball after being regulated by the electrostatic field.
 - 10)In Figure 2c, It is not indicated in the legend about the 77K-14DAP/Ethanol-P, the excitation wavelength, delayed measurement time for phosphorescence, and the insert information for the image.
 - 11)Authors need to be consistent in the use of copolymerization and co-pyrolysis.

Reviewer #3 (Remarks to the Author):

In this manuscript the authors describe their strategies to process urea-formaldehyde resin so aromatic diamines, guest molecules, are incorporated into the backbone of the polymer to yield emission from the diamine moieties. The results are unique, significant and interesting.

To be scientifically accurate the authors need to report the phosphorescence and afterglow of their guest molecules themselves, or references who has measured those before.

In addition, how different are the results if the guest molecules are not incorporated into the backbone of the polymer, but only dispersed or dissolved in the polymer?

The lifetimes of the emissions are presented as one number. How were the lifetimes obtained? How uniform are the decays? What is the error in measuring the lifetimes?

The videos are very informative!

They could be made better if the light, camera and the material were kept in place while filming.

The major drawback is that the paper is written awkwardly, which makes it difficult to be sure if I understand all the scientific results they are explaining. It would be easier to review the article if it had been edited before being submitted for review.

Lines 81 to 89, are the authors describing the work presented in this manuscript, or is the citing the literature? Please rewrite the paragraph to make this clear.

The phrases “on-line derivatization or online derivatization” do not read well. Does “on-line” refer to the polymer “chain” or “backbone”? If so I don’t think there is any need to invent a new and confusing phrase to describe this concept.

REVIEWER COMMENTS

Reviewer #1 (Remarks to the Author):

In this work, based on the on-line derivative strategy proposed by the authors, they developed a urea-formaldehyde resin RTP material with ultra-long afterglow, good stability and excellent processability. The phosphorescence mechanism of UF-RTPs was also investigated. Interestingly, electrostatic field regulation of RTP is achieved by simply mixing RTP microspheres with different electrostatic strengths. The results in this manuscript are of great interest to readers in this field. Therefore, the manuscript could be published after minor revisions.

1. The drawing in Figure 3c is not standard, and the arrow pointing should be clear.

Reply: Thank you very much for your suggestions. According to your suggestion, we have modified Fig. 3c in the revised manuscript. Thanks again for your careful work on our manuscript.

2. Figure 4b should add the label of phosphorescence decay lifetime.

Reply: Thank you for pointing out this problem. We have added the label for phosphorescence decay lifetime in the revised manuscript (Fig. 4b.). Thanks again for this helpful comment.

3. Why use 310 nm UV as excitation source in the photophysical test of UF-RTPs? “As shown in Fig. 2c, 14DAP/UF, 13DAP/UF, and 12DAP/UF all exhibited photoluminescence property, producing blue emission under 310 nm UV excitation.”

Reply: Thanks for your professional suggestions. According to your suggestion, the excitation spectra of UF-RTPs were collected and added in the revised supplementary information as Fig. S22, it was found that all UF-RTPs exhibited significant phosphorescence emission intensity under 310nm excitation. To facilitate characterization of photophysical properties of UF-RTPs, we uniformly took 310 nm as the excitation wavelength. Detailed discussions have been added in the revised manuscript. Thanks again for your consideration on our manuscript.

4. UF-RTPs prepared by pyrolysis showed two efficient phosphorescence emission mechanisms. Is there any synergistic effect between them?

Reply: Thank you very much for your suggestions. As you pointed out, the RTP of UF-RTPs was markedly enhanced by the synergistic effect of the two emission mechanisms. The photophysical

properties of benzimidazole and the other *o*-phenylenediamine derivatives were examined (Supplementary Fig. S27). 910DAPT/UF ($\tau_p=3326$ ms) showed a better phosphor decay life than PMZ/UF ($\tau_p=2815$ ms), while 23DAN/UF and NMZ/UF exhibited a similar result. These results indicated that the synergistic effect of the two luminescence mechanisms improved the RTP performance and prolonged the decay lifetime of the materials. More discussion on this issue was added in the revised manuscript. Thanks again for your consideration on our manuscript.

5. Phosphorescence induced by electrostatic interaction between different microspheres is a very interesting method for enhancing RTP. The adjustment performance is achieved by mixing microspheres with different electrostatic strengths. However, can directly applied electrostatic fields regulate phosphorescence properties on RTP microspheres? “When m0% μ UF: m1% μ UF increased from 0 to 2, the τ_p of the mixture increased from 27 ms to 123 ms (Fig. 5f).”

Reply: Thanks for your valuable comment. As you suggested, we attempted to use an external electrostatic field to regulate the RTP properties of the materials. Accordingly, we constructed an electric field as shown in Fig. R1a, a sample of 9-aminofifi /UF microspheres (9AP/ μ UF) was prepared and placed in the electric field. Surprisingly, the afterglow lifetime of 9AP/ μ UF extended from 3.1 s to 4.1 s before and after electric field activation (Fig. R1b), demonstrating that an external electric field can indeed adjust the photophysical properties of organic RTP materials. The related discussion was added in the revised manuscript. Thanks again for your careful work on our manuscript.

Fig. R1. (a) The applied electric field experimental device. (b) the regulation of the afterglow duration of 9AP/ μ UF by an electrostatic field.

6. Some typical literature about long-lived RTP should be cited, such as [10.1021/jacs.2c02076](https://doi.org/10.1021/jacs.2c02076), [10.1002/smll.202201223](https://doi.org/10.1002/smll.202201223), [10.1002/ange.202203254](https://doi.org/10.1002/ange.202203254), [10.1002/adma.202204415](https://doi.org/10.1002/adma.202204415), [10.1038/s41467-023-43133-1](https://doi.org/10.1038/s41467-023-43133-1), [10.1016/j.chempr.2023.05.023](https://doi.org/10.1016/j.chempr.2023.05.023), etc.

Reply: Thank you very much for your professional suggestions, we have supplemented these references in the revised manuscript (Reference 10, 19, 21, 35, 36, 37). Thanks again for your careful work on our manuscript.

Reviewer #2 (Remarks to the Author):

In this paper, Chen et al. reported a series of ultra-long afterglow UF-RTPs constructed based on the on-line derivatization strategy, which show a series of attractive properties. It should note that “the polymerization of aromatic diamines with urea and formaldehyde” is NOT a copolymerization, conceptually. In addition, the characterization of the samples are insufficient, only IR spectra presented. The monomer residues in the final polymers should be given, and the self-assembly to microspheres should be characterized, and provides reasonable explanations. Overall, this paper could be accepted once the following issues are well addressed.

Reply: Thank you very much for your valuable comment, we have corrected the expression " the polymerization of aromatic diamines with urea and formaldehyde" to "polycondensation of aromatic diamines with urea and formaldehyde" to illustrate the accurate chemical reaction process.

According to your suggestion, solid-state nuclear magnetic resonance (SSNMR) and gel permeation chromatography (GPC) spectra of UF-RTP materials were added in the revised supplementary information. Furthermore, in the revised manuscript, both X-ray diffraction spectroscopy (XRD) and X-ray photoelectron spectroscopy (XPS) were employed to describe the co-polycondensation reaction in detail, and further confirmed the structures of UF-RTP.

As pointed out by the reviewer, there will be residues of paraformaldehyde, urea and guest molecules and their derivatives in UF-RTPs. Thus, UF-RTPs were extracted and the extracts were analyzed by LC-MS (Supplementary Fig. S16-18 and Table S1). A detailed discussion was given on Page 7 of the revised manuscript.

The morphology and particle size of μ UFs were characterized by SEM (Supplementary Fig. S33), and it was found that 0% μ UF-5% μ UF had regular spherical structures, their diameters were about 1-2 μ m (Supplementary Fig. S34). The related discussion on the microsphere morphology has been added to the revised manuscript. Thanks again for your helpful comments.

1. In Figure 1, The benzimidazole is exhibited in the polymer chain by physical blending not chemical bond connections, so what would be the difference if the benzimidazole molecules were directly physically doped? In addition, this fig is full of many phrases, which one is the most important?

Reply:

- (1) Thanks to your valuable advice. As you suggested, a mixture of benzimidazole (BMZ) and UF powders were ground in ethanol to yield UF+BMZ. Photoluminescence spectra and phosphorescence spectra of UF and UF+BMZ were collected under the excitation of 310nm (Fig. R2a, b). Unfortunately, no significant afterglow of UF+BMZ was observed. In the phosphorescence spectra of UF and UF+BMZ (Fig. R2b), the phosphorescence peaks of both were at 461 nm, which was attributed to UF. However, BMZ/UF prepared based on on-line derivatization strategy showed excellent RTP performance with τ of 786 ms (Fig. R2c, d). Therefore, effective phosphorescence emission can be achieved through online derivation strategy. Simple physical blending is difficult to achieve uniform dispersion and effective limiting of guest molecules in UF matrix, and it is impossible to bond in polymer chains, which cannot effectively inhibit non-radiative transitions. Thank you very much for your valuable advice.
- (2) As you suggested, the phrases in Fig.1 have been simplified to highlight the point. The modified figure was added to the revised manuscript as Fig. 1. Thanks again for your careful work on our manuscript.

Fig. R2. (a) Photoluminescence and (b) phosphorescence spectra of UF, UF+BMZ and (c) BMZ/UF. (d) Phosphorescence decay lifetime of BMZ/UF ($\tau=786$ ms).

2. The characterization of UF-RTPs' structure needs to be further clarified, e.g. NMR results and molecular weight data need to be given.

Reply: Thank you very much for your professional suggestions. According to your opinion, the ^{13}C -SSNMR and GPC spectra of the commercial UF, the prepared UF, and 14DAP/UF were collected and added in the revised supplementary information (Fig. S8-13). Detailed discussion has been supplemented in the revised manuscript (Page 5). Thanks again for your careful work.

3. In the photophysical properties of UF-RTPs part, figure 2c, the authors need to further explain why the phosphorescence wavelengths are different in different temperatures, and why there are also two emission peaks in the phosphorescence spectra of 298K-14DAP/UF-P?

Reply:

(1) Thanks for pointing out this problem. We feel very sorry for this mistake. Upon re-examination of the data, we found that "298K-13DAP/UF-P" in Figure 2c of the original manuscript had erroneously placed the phosphorescence spectrum of a high-concentration 13DAP/UF sample. The modified 298K-13DAP/UF-P was shown in the revised manuscript as Fig. 2c, which has significant overlap with the 77K-13DAP/UF-P.

(2) In the phosphorescence spectrum of "298K-14DAP/UF-P," partial fluorescence was detected due to the absence of appropriate gating in the Edinburgh FLS1000 spectrometer. The same problem also occurred in original supplementary information Fig. S19a. Thus, the phosphorescence spectra of 298K-14DAP/UF-P and 910DAPT/UF-P were re-collected by a spectrometer with appropriate gating and added in revised manuscript as Fig. 2c. and Supplementary Fig. S27a, respectively. We hope that the corrections will meet with approval. Thanks again for your helpful comments.

4. the authors need to further explain in Figure 2d, why the phosphorescence lifetime of UF-RTPs is first increased and then decreased with the dopant ratio of guest molecule from 0.1wt%-1wt%?

Reply: Thanks for this valuable comment. Aggregation-induced quenching remains an urgent issue to be addressed in the field of luminous materials (*Chemical Engineering Journal* **469**, 143929(2023); *Chemical Communications* **58(22)**, 3641-3644(2022)). In UF-RTPs, guest molecules exist in the matrix as free molecules before reaching the critical concentration, without aggregation. At this stage, the phosphor quantum yield (Φ_p) and the phosphor decay lifetime (τ_p) are improved with the increase of the concentration of phosphor guest molecule. At the critical concentration, Φ_p and τ_p achieve the maximum. However, as the concentration of guest molecules further increases, guest molecules will aggregate due to the strong π - π interaction. The

excited state energy will be dissipated through non-radiative ways, thus accelerating the quenching of triplet excitons, resulting in the reduction of Φ_p and τ_p . Thanks again for your careful work on our manuscript.

5. In Figure 4c, the authors demonstrate the phosphorescent lifetime of the polymer after 6 months of storage, when the polymer is placed in water and then dried, how does the phosphorescent lifetime change?

Reply: Thanks to your professional advice. According to your suggestion, the phosphorescence decay lifetime of 910DAPT/UF after soaking in water and vacuum drying at 90°C was collected. It was found that treated 910DAPT/UF still exhibited good phosphorescence lifetime without significant decrease (Fig. R3), indicating that the UF-RTPs prepared through the on-line derivatization strategy displayed good structural stability. The discussion about water resistance of 910DAPT/UF has been added to our revised manuscript (Page 11). Thanks again for this helpful comment.

Fig. R3. (a) The phosphor decay life of 910DAPT/UF during (b) four cycles of washing and drying.

6. Apart from the phosphorescent lifetime, the quantum yield of phosphorescence and fluorescence needs to be given, and the difference in the strength and lifetimes of the polymer in noble gas and air should be further compared.

Reply: Thanks for your professional advice. According to your opinion, the luminescent quantum yield (Φ_{pl}) and phosphorescent quantum yield (Φ_p) of UF-RTPs have been supplemented (Supplementary Fig. S23). A detailed discussion on the photophysical properties of UF-RTPs was added in the revised manuscript (Page 7). Subsequently, we further compared the

changes in phosphorescent intensity (Supplementary Fig. S29) and phosphorescence decay lifetime of UF-RTP (Supplementary Fig. S30) in N₂ or air. The phosphorescence intensity of the UF-RTPs in air was slightly lower than that of UF-RTP in N₂. It is presumed to be due to the absorption of O₂ by UF-RTPs, resulting in the quenching of triplet excitons and a decrease in phosphorescence emission intensity. Similarly, the phosphorescence decay lifetime of UF-RTPs in air was slightly shorter than that of UF-RTPs in N₂ environment. A detailed discussion of stability of UF-RTPs has been added to the revised manuscript (Page 11). Thanks again for this helpful comment.

7. More experiments and evidence are needed to elucidate the effect of dielectric property of the mixture and the polarization of guest molecules to its RTP performance. And how to explain the decreased lifetime with the higher mass ratio of 0% μ UF and 1% μ UF.

Reply:

- (1) Thank you very much for your valuable comments. Studies on the generation and distribution of electrostatic charges in organic polymers have shown that the greater the difference in dielectric constants of materials, the easier it is for electrostatic effects to occur. To verify this, six polymers were selected based on their polarity and quantum chemical computations were performed. Here, Gaussian 09 was used for structural optimization at B3LYP/6-31G** level, then Multiwfn 3.8 was used to calculate the molecular surface electrostatic potential, and Molecular Polarity Index (MPI) was calculated. MPI was taken to assess the polarity of polymer molecules. The results indicated that UF presents the largest molecular polarity with an MPI index of 22.04, while polystyrene has the lowest molecular polarity with an MPI of 6.8 (Fig. R4). Based on the guidance of theoretical calculations, polystyrene microspheres (μ PS) were prepared by dispersion polymerization without adding guest molecules. By mixing μ PS with 1% μ UF, the phosphorescence performance of 1% μ UF could be adjusted from 26 ms to 41 ms (Fig. R5c). Therefore, it found that by mixing two different polarities materials, the dielectric constant of the mixture could be adjusted to enhance the electrostatic effect, thus regulating the RTP performance. This is consistent with the trend of dielectric constant the 0% μ UF and 1% μ UF mixtures in the revised manuscript (Page 12, Supplementary Fig. S37b).
- (2) As you mentioned, the higher the mass ratio between 0% μ UF and 1% μ UF, the shorter the phosphor decay life. With the increase of mass ratio between 0% μ UF and 1% μ UF, the concentration of guest molecule in the complex further decreases, resulting in the reduction of luminescence intensity and phosphor lifetime. Thanks again for this helpful comment.

Fig. R4. Optimized conformation and electrostatic potential distribution of polymer molecular fragments.

Fig. R5. (a) photoluminescence spectra, (b) phosphorescence spectra and (c) phosphorescence decay lifetime spectra of a series of complexes of μ PS and 1% μ UF.

8. Carefully check the whole manuscript. There are some mistakes, for example, Ref. 28 citations are formatted incorrectly; RNH3 and R2NH2 are the wrong expressions.

Reply: Thank you very much for pointing out the error in our manuscript. We feel sorry for our carelessness. We have carefully checked the manuscript and corrected the reference format and expression errors. The specific changes are marked in the revised manuscript. We appreciate your careful work to improve the quality of our manuscript.

9. In Figure 1, there is no appearance change shown in the schematic diagram of the ball after being regulated by the electrostatic field.

Reply: Thank you very much for your suggestions, we have supplemented the electrostatic strength diagram in the revised manuscript Fig.1. Thanks again for your careful work on our manuscript.

10. In Figure 2c, It is not indicated in the legend about the 77K-14DAP/Ethanol-P, the excitation wavelength, delayed measurement time for phosphorescence, and the insert information for the image.

Reply: Thank you very much for your valuable suggestion. According to your opinion, the detailed discussion about Fig. 2c was supplemented in the revised manuscript (page 6, 7). Thanks again for your careful work on our manuscript.

11. Authors need to be consistent in the use of copolymerization and co-pyrolysis.

Reply: Thanks for your suggestion, we have unified the expressions with "polycondensation" in the revised manuscript. The revised phrases have been marked in the revised manuscript. Thanks again for your careful work.

Reviewer #3 (Remarks to the Author):

In this manuscript the authors describe their strategies to process urea-formaldehyde resin so aromatic diamines, guest molecules, are incorporated into the backbone of the polymer to yield emission from the diamine moieties. The results are unique, significant and interesting.

1. To be scientifically accurate the authors need to report the phosphorescence and afterglow of their guest molecules themselves, or references who has measured those before.

Reply: Thanks for your valuable suggestion. Considering that the aromatic diamine guest molecules can emit phosphorescence only when the molecular thermal motion is inhibited at low temperature (77K), we collected 77K phosphorescence spectra of guest molecules (Fig. 2c) according to your suggestion, including 77K-14DAP/Ethanol-P, 77K-14DAP/UF-P, 77K-13DAP/Ethanol-P, 77K-13DAP/UF-P, 77K-12DAP/Ethanol-P, and 77K-12DAP/UF-P. And the intrinsic photophysical properties of the guest molecules were compared. Detailed discussions of the photophysical property of UF-RTPs and guest molecules were supplemented in the revised manuscript (Page 5). Thanks again for your careful work on our manuscript.

2. In addition, how different are the results if the guest molecules are not incorporated into the backbone of the polymer, but only dispersed or dissolved in the polymer?

Reply: Thank you for your professional suggestion. According to your suggestion, UF+BMZ was obtain by grounding BMZ and UF powders in ethanol. Photoluminescence spectra and phosphorescence spectra of UF and UF+BMZ were collected under the excitation of 310nm (Fig. R2a, b). However, no significant afterglow of UF+BMZ was observed. In the phosphorescence spectra of UF and UF+BMZ (Fig. R2b), the phosphorescence peaks of both were at 461 nm, which was attributed to UF. It was worth noting that BMZ/UF prepared based on on-line derivatization strategy showed excellent RTP performance (Fig. R2c, d). Therefore, we speculate that simple physical blending is difficult to achieve uniform dispersion and effective limiting of guest molecules in UF matrix, and it is impossible to bond in polymer chains, which cannot effectively inhibit non-radiative transitions. Thanks again for your careful work.

3. The lifetimes of the emissions are presented as one number. How were the lifetimes obtained? How uniform are the decays? What is the error in measuring the lifetimes?

Reply:

(1) Thank you for your suggestion. Phosphorescence decay lifetime refers to the time taken for the phosphorescence to decay to 1/e of its maximum intensity after the material is excited.

The phosphorescence decay lifetime spectra were collected by the photon counting method of Edinburgh FLS1000. And *Fluoracle* software was used to fit to obtain the phosphorescence decay lifetime. The phosphorescence lifetime fitting process of UF-RTPs was shown in Fig. R7.

- (2) The phosphorescence decay process typically follows an exponential decay function which was modeled as the sum of multiple exponential components, as shown in Eq. (1).

$$I(t) = \sum_{i=1}^n B_i \exp\left(\frac{-t}{\tau_i}\right) \quad \text{Eq. (1)}$$

Here $I(t)$ stands for the measured phosphorescence intensity and is a function of time t ; τ_i are the individual exponential components; and B_i are the coefficients of each exponential term.

- (3) According to your advice, the phosphorescence decay lifetime of 0.1wt%-14DAP/UF (Fig. R6) was tested three times, and the results of the three tests were 258ms, 259ms, and 260ms with an error of approximately 1 ms (0.38%).

Fig. R6. Phosphorescence decay lifetime of 0.1wt%-14DAP/UF.

Fig. R7. Fitting curves of (a)-(k) UF-RTPs phosphorescence decay lifetime.

4. The videos are very informative! They could be made better if the light, camera and the material were kept in place while filming.

Reply: Thank you very much for your valuable advice. We have fixed the camera and light source in suitable positions and re-photographed UF-RTPs powders. The revised video has been uploaded to the supplementary videos. Thanks again for your careful work on our manuscript.

5. The major drawback is that the paper is written awkwardly, which makes it difficult to be sure if I understand all the scientific results they are explaining. It would be easier to review the article if it had been edited before being submitted for review.

Reply: Thank you very much for your valuable suggestions regarding our manuscript. We apologize for our poor language skills. Therefore, we have carefully revised the language and corrected any inappropriate expressions according to the editor's and reviewers' comments. All revisions have been marked in the revised manuscript. In this revised manuscript, we have reported a urea-formaldehyde resin-based RTP material (UF-RTPs) with excellent RTP performance and stability. UF-RTPs achieved a maximum afterglow duration of 47 s and a phosphorescent decay lifetime of 3.3 s. Additionally, UF-RTPs can be prepared into regular microspheres by dispersion polymerization. By adjusting the content of guest molecules, a series of organic RTP microspheres with different electrostatic potential strengths can be obtained (0% μ UF-5% μ UF). RTP decay life was increased from 27 ms to 123 ms by adjusting the electrostatic potential strength of the complexes with different mass ratios of 0% μ UF and 1% μ UF. This is also the first report on the regulation of RTP materials using electrostatic field. Thanks again for this helpful comment.

6. Lines 81 to 89, are the authors describing the work presented in this manuscript, or is the citing the literature? Please rewrite the paragraph to make this clear.

Reply: Thanks for your guidance. Sorry for the inconvenience to your review caused by our poor language expression. Lines 81-89 summarize the work of our manuscript. According to your suggestion, we have rewritten the paragraph to clearly express our work. Thanks again for your careful work on our manuscript.

7. The phrases "on-line derivatization or online derivatization" do not read well. Does "on-line" refer to the polymer "chain" or "backbone"? If so I don't think there is any need to invent a new and confusing phrase to describe this concept.

Reply: Thank you very much for pointing out the mistakes on our manuscript. We are sorry for our carelessness. In the revised manuscript, the phrase of "on-line derivatization" has been unified to ensure the consistency of expression. We appreciate your careful work.

REVIEWERS' COMMENTS

Reviewer #1 (Remarks to the Author):

The authors have revised the manuscript fully and it is suitable for publication as it is.

Reviewer #2 (Remarks to the Author):

The authors have well responded all concerns raised by all reviewers, The manuscript was well revised according to the comments and this reviewer believes that the quality of this paper can be published in NC.

Reviewer #3 (Remarks to the Author):

The results described in this manuscript are noteworthy, and they should be published, the videos and the figures make the results believable. However, before this manuscript is published in any journal, the author have to address the following:

The introduction of the manuscript is not written so the broad readers of this journal can appreciate it. There is too much jargon, that are not clearly defined and make it hard to appreciate the science described in the manuscript. In addition, the manuscript is written in awkward English that needs to be edited carefully to make it readable to the general reader.

For example, in the introduction I don't have a good insight into what the authors mean

Page 2, line 40. I don't know what is "host-guest strategy and doping strategy." Do the authors mean host-guest complexes and doping of polymers?

Please define specifically what is doping strategy and why it is simple.

What do the authors mean by "flexible formability and strong versatility", it reads like non-specific jargon.

Page 3. Define τ_p

The authors have not defined what they mean by saying "derived on-line". Please clearly.

Also define “intelligent dynamic regulations “,

Results

Figure 2. it is difficult comprehend what Figure 1 displays, the labels at the top are not legible.

Figure 4. How different are these lifetimes, is there a difference between 3326 ms versus 3269 ms. Does the data support the listed accuracy of these lifetimes? Should the be listed as 3.3×10^3 ms?

Page 11, line 226. The phrase “As we all know” does not belong in any scientific manuscript.

The Figures have improved, and the videos nicely support the finding.

The manuscript could be further enhanced by adding a short conclusion that puts the results obtain in the work in perspective to the field and explain why they are important.

I am disappointed that the authors have not made the manuscript easy to read, because the results are interesting.

Reviewer #1 (Remarks to the Author):

1. The authors have revised the manuscript fully and it is suitable for publication as it is.

Reply: Many thanks for your kind recommendation!

Reviewer #2 (Remarks to the Author):

1. The authors have well responded all concerns raised by all reviewers, The manuscript was well revised according to the comments and this reviewer believes that the quality of this paper can be published in NC.

Reply: Thanks for your kind recommendation and careful work for this manuscript!

Reviewer #3 (Remarks to the Author):

The results described in this manuscript are noteworthy, and they should be published, the videos and the figures make the results believable. However, before this manuscript is published in any journal, the author have to address the following:

The introduction of the manuscript is not written so the broad readers of this journal can appreciate it. There is too much jargon, that are not clearly defined and make it hard to appreciate the science described in the manuscript. In addition, the manuscript is written in awkward English that needs to be edited carefully to make it readable to the general reader. For example, in the introduction I don't have a good insight into what the authors mean

Reply: Thank you for your professional advice. We are sorry for our poor language skills. As your suggestion, the terminologies and incorrect expressions have been supplemented, improved and marked in the revised manuscript. In addition, we have checked and improved the language in the manuscript in the hope that readers will understand our scientific views.

1. Page 2, line 40. I don't know what is "host-guest strategy and doping strategy." Do the authors mean host-guest complexes and doping of polymers?

Reply: Thanks for your guidance. We are sorry for the inconvenience caused to your review due to our poor language expression. The "host-guest strategy" refers to the realization of

phosphorescence emission through the inclusion of phosphorescent guest molecules by host molecules. “Doping strategy” refers to the doping of phosphorescent guest molecules into a rigid matrix to obtain RTP materials (*Nature Reviews Materials* **5** (2020): 869-885).

According to your suggestions, in the revised manuscript we adjusted the expression of these two strategies to “host-guest complexation and matrix doping strategy”. Thanks for your careful work.

2. Please define specifically what is doping strategy and why it is simple.

Reply: Thanks for your suggestion. In the matrix doping strategy, the phosphorescent guest molecules are doped in the rigid matrix to achieve high performance RTP materials. The matrix can effectively promote the emission of phosphorescence through suppression of non-radiative transitions and the isolation of quenching factors (*Nature Communications* **12** (2021): 2297).

Compared with molecular engineering, crystal engineering and host-guest complexation, matrix doping strategy can simplify the preparation process. It shows the characteristics of easy preparation and good RTP performance.

According to your suggestion, we have added a discussion of “matrix doping strategy” in the revised manuscript, so as to enhance the readability of the manuscript. Thanks again for your careful work.

3. What do the authors mean by “flexible formability and strong versatility”, it reads like non-specific jargon.

Reply: Thank you for your comment. RTP materials prepared based on the matrix doping strategy showed good processing performance. For example, choosing polyvinyl alcohol or polyacrylic acid as matrix, it can be processed into phosphorescent film; bulk phosphorescent material can be prepared by vulcanization process with urea-formaldehyde resin as matrix. It is based on the above RTP materials can be processed into a variety of shapes and have a variety of functions, which can be widely used in chemical sensing, advanced anti-counterfeiting encryption and lighting fields.

According to your suggestion, we use “processing performance and multi-functionality” instead of “flexible formability and strong versatility” in the revised manuscript to enhance the readability of the manuscript.

4. Page 3. Define τ_p . The authors have not defined what they mean by saying “derived on-line”. Please clearly. Also define “intelligent dynamic regulations “,

Reply: Thank you very much for your helpful suggestions, we apologize for our poor writing skills. We have added the definition of τ_p on Page 3 of the revised manuscript. The description of

“on-line derivation” and “intelligent dynamic response” have been added on Page 3 of the revised manuscript. Thank you again for your careful work.

Results

5. Figure 2. it is difficult comprehend what Figure 1 displays, the labels at the top are not legible.

Reply: We feel sorry for the inconvenience caused to your review due to the lack of clarity in the picture, and we have re-adjusted Figure 2a to make it more readable. Thanks for your careful work.

6. Figure 4. How different are these lifetimes, is there a difference between 3326 ms versus 3269 ms. Does the data support the listed accuracy of these lifetimes? Should the be listed as 3.3×10^3 ms?

Reply: Thanks for your suggestion, Figure 4c shows the variation of phosphor decay life of 910DAPT/UF in air with time. The fitting lifetime of 910DAPT/UF-0 month and 910DAPT/UF-1 month were 3326 ms and 3269 ms respectively, and the fitting lifetime of 910DAPT/UF-1 month was slightly decreased compared with 910DAPT/UF-0 month. In addition, in Figure 4c, the phosphor decay curve of 910DAPT/UF-1 month also showed a slight downward shift compared with 910DAPT/UF-0 month, which was consistent with the results of fitting lifetime results. Therefore, the use of 3326 ms and 3269 ms can more accurately and reliably demonstrated the results. Thank you again for your careful work.

7. Page 11, line 226. The phrase “As we all know” does not belong in any scientific manuscript.

Reply: Thank you for your professional advice. We are sorry for our poor language expression, and we have revised the incorrect expression in the revised manuscript.

8. The Figures have improved, and the videos nicely support the finding.

Reply: Thank you for your helpful advice for our manuscript.

9. The manuscript could be further enhanced by adding a short conclusion that puts the results obtain in the work in perspective to the field and explain why they are important.

Reply: Thank you very much for your professional advice. In the section "Construction of microspheres for tunable RTP by electrostatic field", we have added a brief summary on Page 13 of the revised manuscript, highlighting the application prospects and importance of electrostatic field response RTP materials. Thanks again for your helpful comment.

10. I am disappointed that the authors have not made the manuscript easy to read, because the results are interesting.

Reply: We are sorry for our poor expression. According to the reviewer's suggestions, we have further modified and improved the manuscript, and hope to get your approval.